# Leaf Area Estimation by Photographing Leaves Sandwiched between Transparent Clear File Folder Sheets

Kohei Koyama

Faculty of Education, Hokkaido University of Education, Asahikawa Campus, Hokumoncho 9, Asahikawa 070-8621, Japan; koyama.kohei@a.hokkyodai.ac.jp

**Abstract:** Image analysis is a promising method for in situ leaf area measurement. However, as leaves are three-dimensional, the use of two-dimensional images captured using a digital camera can result in underestimation. To overcome this problem, we tested a clear folder method. Before photographing leaves with a digital camera, we flattened the leaves by sandwiching them between a pair of transparent plastic clear file folder sheets, which are stationery implements for carrying documents. Although similar methods have been previously proposed, their applicability to species with different leaf shapes has never been investigated. We tested the efficacy of this method using 12 species from various taxa (monocots, magnoliids, and basal and core eudicots) and leaf morphology (entire vs. lobed, simple vs. compound leaves, small and large leaves). Individual leaf areas and the Montgomery parameters obtained using this method were then compared with those obtained using the standard method, which employs a flatbed digital scanner. We observed strong correlations ($R^2 > 0.98$) between the camera and scanner data. The regression slopes were close to unity (0.96–1.01) and the intercepts were close to zero. These findings suggest that the clear folder method can be used as an inexpensive alternative method to estimate the area of leaves in situ with acceptable accuracy. An introductory manual for readers unfamiliar with image analysis using ImageJ is presented in the end of the paper.

**Keywords:** leaf area; image analysis; clear folder method; digital camera; in situ; leaf size; Montgomery equation; length-times-width equation; non-destructive; Montgomery parameter

## 1. Introduction

Leaf area is a major determinant of whole-plant photosynthesis [1–3], transpiration [4,5], and respiration [6–8]. It also determines within- and between plant shading and consequent competition [9–11]. At the ecosystem level, the stand leaf area or leaf area index (LAI; i.e., total leaf area per unit land area) determines the ecosystem carbon uptake rate [12–14], ecosystem transpiration [15,16], and $CH_4$ emission [17], and substantially affects environmental factors such as soil temperature [18,19]. In agriculture, leaf area has been extensively studied because it is a major determinant of crop yields [20–22]. Measurement of leaf area has thus become one of the most important topics in plant science.

The Montgomery equation (ME) [23,24], also known as the leaf length-times-width equation [25], is an established empirical relation that states that within each species, the area of an individual leaf is proportional to the product of its lamina length and width. The main advantage of the ME is that it allows for non-destructive estimation of leaf area through the simple measurement of leaf length and width in situ. The ME has been widely used to estimate the leaf areas of various plant species, including woody plants [26,27], herbaceous plants [26,28,29], lianas [26,30], monocots [26,31,32], Magnoliids [25,26], ferns [26], and important crop species such as maize [33], pear [34], and grapes [35,36]. However, to apply the ME to estimate leaf area, a species-specific proportionality constant, known as the Montgomery parameter, must be determined in advance.

This can be achieved by fitting a regression line using small number of leaf samples. Therefore, when we use the ME, an independent measurement of leaf area using a different method is needed to estimate the Montgomery parameter.

Recently, digital camera image analyses have been used to estimate individual leaf area [37–50], whole-plant leaf area or growth rate [42,51–57], leaf area index [52,58–63], leaf morphology [64], leaf color [65], and leaf angle distribution [66]. However, a major limitation of these approaches is that photographs are only a two-dimensional (2D) projection of leaves, whereas leaves (or groups of leaves on a stem) are three-dimensional (3D) (i.e., folded, curled, or wiggled) [67]. The 3D nature of leaves and foliage plays a crucial ecological function in controlling light flow and temperature on the leaf surface [68–70]. Since measuring the leaf area using 2D photos results in its underestimation [57], a correction factor must be estimated using standard measurements with destructive sampling. Although this issue has long been acknowledged in image analysis studies (e.g., [55,57]), it is commonly neglected. Notably, recent techniques to reconstruct the 3D surface of a plant using various camera angles or a 3D camera have solved this problem [66,71–77]. Despite their great advantages, these 3D techniques require specific camera setups (e.g., the camera(s) should be fixed at a position in a laboratory [71–73]) or complex image processing [66,76,77], both of which are difficult for field plant scientists in practice. For these reasons, many field plant scientists or agronomists still use simpler approaches such as the length-times-width equation and its variants [26,36] or allometry [25,35,47].

Here, we assessed the applicability of an alternative strategy to overcome the 3D problem. For the proposed method, the leaves were flattened using a pair of transparent plastic sheets commonly available as stationery items (costing less than $1 each). The clear folder method presented here is a simplified version of methods described firstly by Costa et al. [47] and later by Liu et al. [49], Tu et al. [78], and Siswantoro et al. [79], all of which used transparent sheets in a similar manner. However, in these previous studies, only species with leaves of normal size and shape were evaluated. Therefore, the applicability of this method to species with varying leaf shapes (lobed or heavily dissected, exceptionally large or small leaves) remains unclear. The objective of this study was to test the efficacy of this simple, low-cost method of in situ leaf area measurement for species from various taxa and leaf forms.

## 2. Materials and Methods

Additional information regarding the protocol followed can be found in the Appendix A of this article.

### 2.1. Study Species

The study was conducted in September 2022 at the Asahikawa Campus of Hokkaido University of Education (43°47′ N 142°20′ E, altitude: 107 m a.s.l.) in Asahikawa City, Hokkaido, which is located in a cool temperate region in Japan. The mean annual temperature and precipitation in 2013–2022 were 7.61 °C and 1158 mm, respectively, at the Asahikawa Weather Station, which is approximately 4 km from the study site (data from the Japan Meteorological Agency). Twelve species covering various taxa, leaf morphology, and leaf sizes were selected (Table 1). Hereafter, each species is referred to by its genus name (e.g., *Acer*).

### 2.2. Leaf Photography

For each species, we selected five healthy regular-shaped undamaged leaves. For species with compound leaves (*Trifolium* and *Oxalis*), a leaflet was considered equivalent to a leaf [26,80]. In the case of compound leaves, five leaflets were sampled from five different leaves to avoid pseudoreplication. Hereafter, we use the word "leaf" to refer to either a leaf or leaflet. Each leaf lamina (i.e., leaf blade) was sandwiched in a clear file folder (310 mm × 220 mm; 737-SP, King Jim, Tokyo, Japan) and placed on

a white clipboard. A clear file folder (also called "sheet protector", "plastic sleeve", or "clear file") is a pair of transparent plastic sheets used as a stationery implement to carry documents (Figure 1a). Variants of such folders are available worldwide and are suitable for this method. This sandwiching step was performed to mimic a standard leaf area meter (Li-1300; LI-COR, Lincoln, NE, USA), which sandwiches leaves using a belt conveyor. Weights (pruning scissors) were placed near (but not directly on) the leaf to gently press it without causing damage. A paper card (30 mm × 68 mm, Tan-101-P, KOKUYO, Tokyo, Japan) was sandwiched together with each leaf and used as a 3 cm scale bar (Figures 1 and 2).

**Table 1.** List of plant species.

| Code | Species | Common Name | Higher Taxonomy | Order |
|---|---|---|---|---|
| 1. Acpi | *Acer pictum* Thunb. | Maple | Core eudicots | Sapindales |
| 2. ArtX | *Artemisia* sp.[*1] | - | Core eudicots | Asterales |
| 3. Comc | *Commelina communis* L. | Asiatic dayflower | Monocots | Commelinales |
| 4. Fasa | *Fallopia sachalinensis* (F.Schmidt) Ronse Decr. | Giant knotweed | Core eudicots | Caryophyllales |
| 5. Hoco | *Houttuynia cordata* Thunb. | Fish mint | Magnoliidae | Piperales |
| 6. HydX | *Hydrocotyle* sp. | Water pennyworts | Core eudicots | Apiales |
| 7. Mgkb | *Magnolia kobus* DC. | Kobushi magnolia | Magnoliidae | Magnoliales |
| 8. Oxco | *Oxalis corniculata* L.[*2] | Creeping woodsorrel | Core eudicots | Oxalidales |
| 9. PlaX | *Platanus* sp. [*3] | Plane tree | Basal eudicots | Proteales |
| 10. Poav | *Polygonum aviculare* L. | Common knotgrass | Core eudicots | Caryophyllales |
| 11. Prsa | *Prunus sargentii* Rehder | Sargent's cherry | Core eudicots | Rosales |
| 12. Trre | *Trifolium repens* L. | White clover | Core eudicots | Fabales |

| Code | T: Tree H: Herb | Simple or Compound | Form of Leaf (or Leaflet) | Area of Individual Leaf or Leaflet (Range) (cm$^2$) | Sample Size [*4] |
|---|---|---|---|---|---|
| 1. Acpi | T | Simple | Lobed, protruding [*5] | 3.601–112.561 | 60 |
| 2. ArtX | H | Simple | Highly dissected | 0.969–28.963 | 55 |
| 3. Comc | H | Simple | Parallel vein | 0.633–29.126 | 55 |
| 4. Fasa | H | Simple | Large, protruding | 74.491–236.997 | 5 |
| 5. Hoco | H | Simple | Entire, cordate | 2.205–80.546 | 60 |
| 6. HydX | H | Simple | Lobed, toothed | 0.167–11.507 | 56 |
| 7. Mgkb | T | Simple | Entire, obovate | 45.900–82.608 | 5 |
| 8. Oxco | H | Compound | Very small, obcordate | 0.073–1.336 | 60 |
| 9. PlaX | T | Simple | Large, lobed, protruding | 4.312–343.651 | 60 |
| 10. Poav | H | Simple | Very small | 0.0584–2.033 | 55 |
| 11. Prsa | T | Simple | Toothed | 31.653–63.937 | 5 |
| 12. Trre | H | Compound | Serrulate | 0.209–5.057 | 60 |

[*1] We tentatively identified this species as *A. montana* (Nakai) Pamp. Because several species that are difficult to distinguish from each other are grouped within the same genus, we identified it at the genus level. [*2] We did not distinguish *O. corniculata* from a hybrid of *O. corniculata* and *O. dillenii* Jacq. [*3] We tentatively identified this species as a hybrid *P.* × *hispanica* (syn. *P.* × *acerifolia* (Aiton) Willd.) based on the morphology of its leaves. Since its classification is controversial, we identified it to the genus level. [*4] For each species, the sample consisted of five leaves for photography and scanning plus 50–55 leaves (for estimation of the regression-based Montgomery parameter). For *Fallopia*, *Magnolia*, and *Prunus*, because the regression-based Montgomery parameters were obtained in a previous study [25], only five leaves per species were sampled in the present experiment (see the main text for details). [*5] Protruding leaves have lamina that extends below their base (sensu [26]).

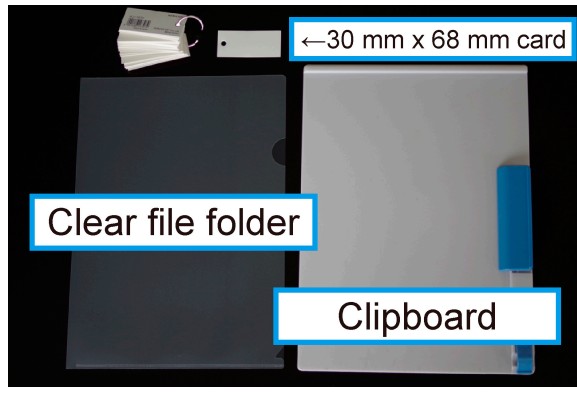

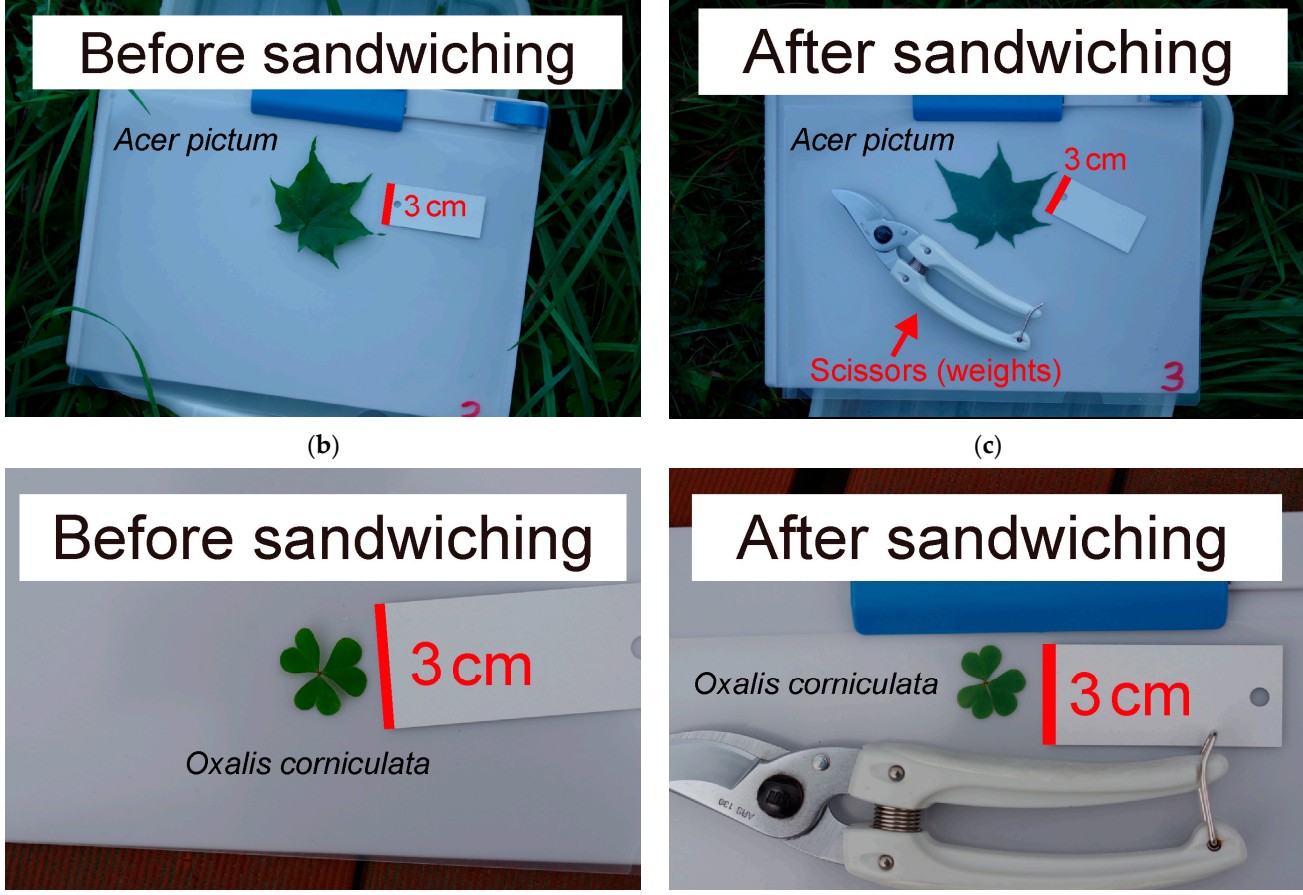

**Figure 1.** Sandwiching leaf (or leaflet) laminas using a transparent clear file folder to flatten the laminas. Top (**a**): a transparent clear folder used in this experiment (310 mm × 220 mm; 737-SP, King Jim, Tokyo, Japan), a white plastic clipboard, and rectangular paper cards (30 mm × 68 mm, Tan-101-P, KOKUYO, Tokyo, Japan). (**b**–**e**) When sandwiching, pruning scissors were placed on a clear folder aside the leaf as weights to gently press the laminas. A rectangular sheet of paper (30 mm × 68 mm) was sandwiched along with the leaf laminas and used as a scale for the image analysis afterward. Note that before sandwiching, the laminas are three-dimensional, whereas after sandwiching the laminas are flat and thus suitable for area measurements by image analysis. Photographs were taken in (**a**) April 2023 and (**b**–**e**) September 2022 by Kohei Koyama. The original high-resolution images are available as Supplementary Materials.

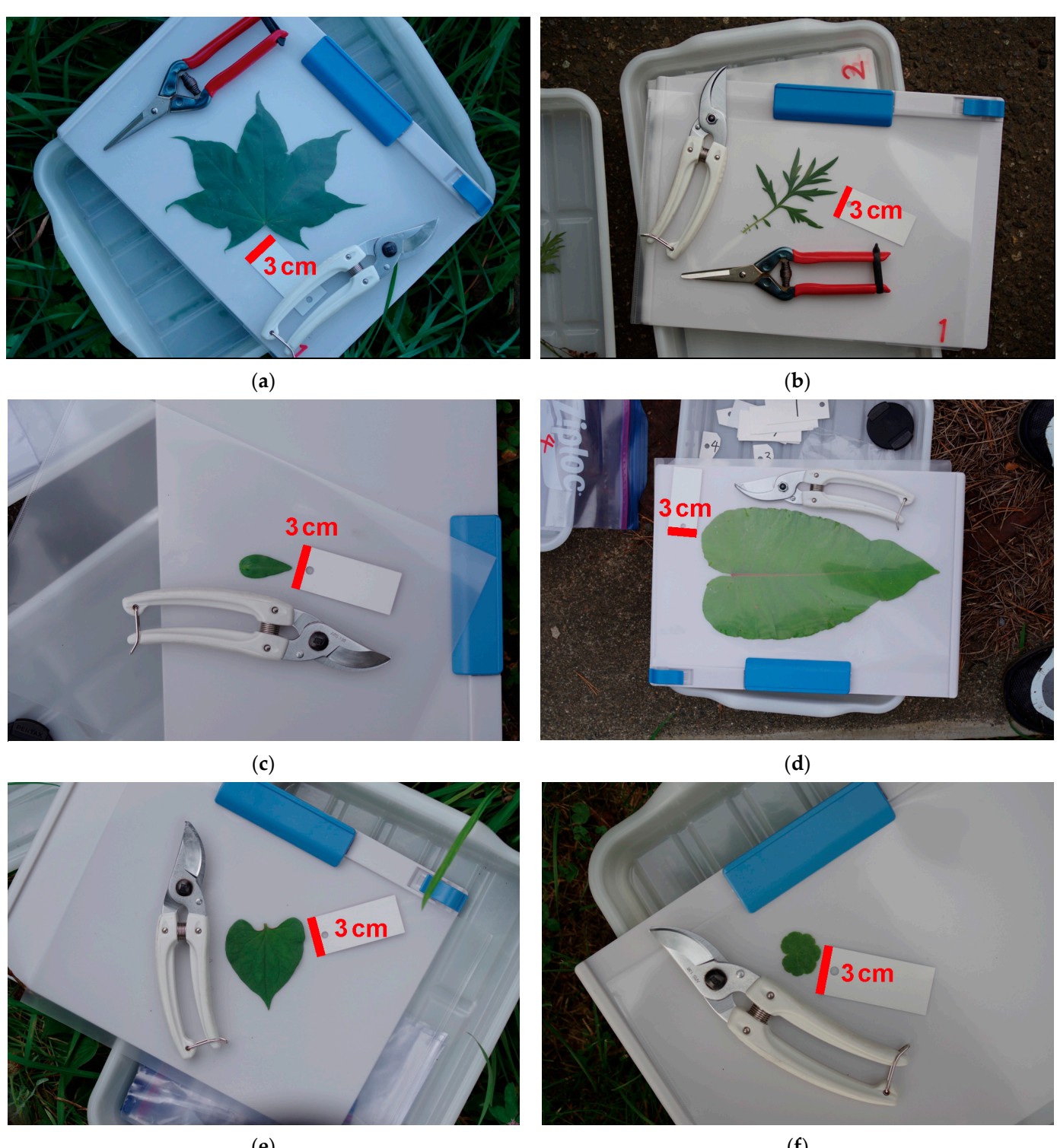

**Figure 2.** *Cont.*

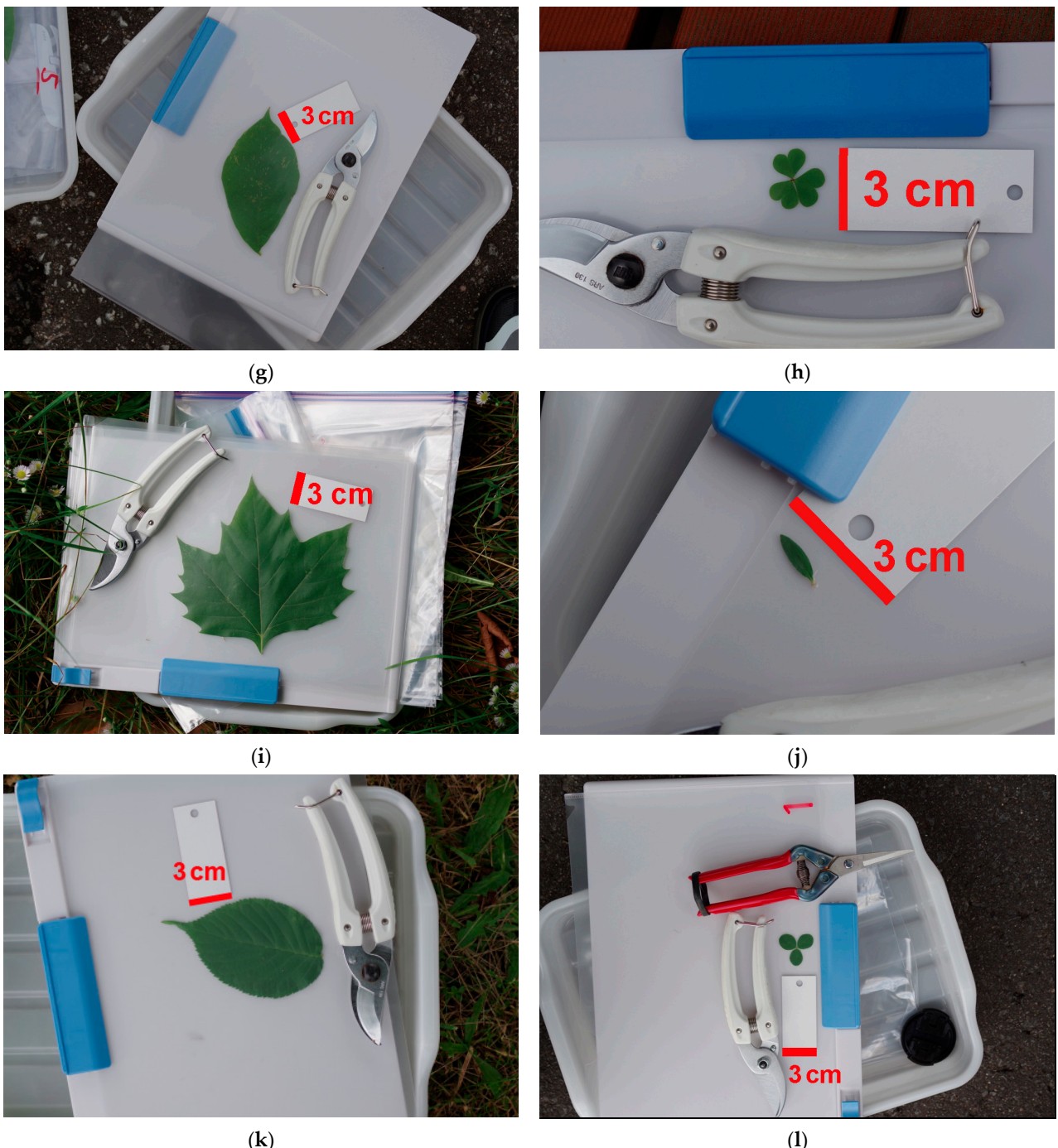

**Figure 2.** Photographing leaves or leaflets of the 12 species in the field. All leaf laminas were sandwiched in a clear file folder, which is a pair of transparent plastic sheets connected at the edges and placed on a white clipboard. A small rectangular sheet of paper (30 mm × 68 mm) was sandwiched together with the laminas and used as a 3 cm scale. (**a**) Maple (*Acer pictum* Thunb.) (**b**) *Artemisia* sp. (**c**) Asiatic dayflower (*Commelina communis* L.). (**d**) Giant knotweed (*Fallopia sachalinensis* (F.Schmidt) Ronse Decr.). (**e**) Fish mint (*Houttuynia cordata* Thunb.). (**f**) Water pennyworts (*Hydrocotyle* sp.). (**g**) Kobushi magnolia (*Magnolia kobus* DC.) (**h**) Creeping woodsorrel (*Oxalis corniculata* L.). (**i**) Plane (*Platanus* sp.). (**j**) Common knotgrass (*Polygonum aviculare* L.). (**k**) Sargent's cherry (*Prunus sargentii* Rehder). (**l**) White clover (*Trifolium repens* L.). Photographs were taken in September 2022 by Kohei Koyama. The original high-resolution images are available as Supplementary Materials.

Each leaf lamina was photographed using a digital camera (PENTAX K-70 and D FA Macro 50 mm F2.8; RICOH Imaging, Tokyo, Japan). No illumination other than natural sunlight was used. All camera settings (e.g., exposure and resolution) were in automatic mode. The leaf lamina was placed at the center of the image to avoid lens distortion. We did not use a tripod because (1) neither the angle nor distance between the lens and leaf affects the results (see Discussion) and (2) the image resolution was sufficiently high without a tripod.

### 2.3. Scanning Leaves with a Flatbed Scanner

We used the methods described in our previous study [25]. Immediately after the photography procedure described above, we sampled the same five leaves per species and stored them in closed wet plastic bags to avoid desiccation. The adaxial surface of the leaf laminas was then scanned using a flatbed digital scanner (400-SCN025, Sanwa Supply, Okayama, Japan) at a resolution of 600 dpi on the same day as sampling.

### 2.4. Leaf Size Measurement

Individual leaf area ($A_{leaf}$) is defined as the area of one side of each leaf lamina. Leaf length ($L_{leaf}$) is defined as the length of the lamina measured from the distal-most point to the point where it joins the petiole (Figure 3). As the petioles of *Artemisia* cannot be separated from the lamina midrib (Figure 1b), their $L_{leaf}$ refers to the full length of the leaf. Leaf width ($W_{leaf}$) is defined as the maximum lamina width perpendicular to the midvein. For images obtained using either a camera or scanner, the length, width, and area of the leaves were measured using ImageJ 1.53a free software [81] (see Appendix A for an introductory manual of leaf area measurement by using ImageJ). Measured values (expressed in dots) were then converted to length in centimeters using measurements of the 3 cm scale.

*Fallopia sachalinensis*

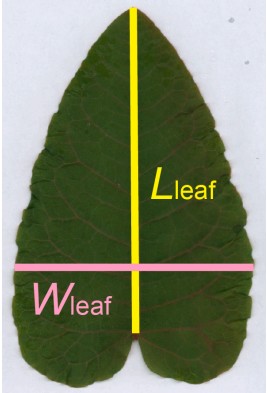

**Figure 3.** Leaf length ($L_{leaf}$) and width ($W_{leaf}$). A leaf lamina of Giant knotweed (*Fallopia sachalinensis* (F.Schmidt) Ronse Decr.) is shown. This photograph was taken in September 2022 by Kohei Koyama.

Binarization (i.e., converting a color image to a black and white image) was performed automatically using the "Make Binary" function in ImageJ (Figure 4; see Appendix A for details). After binarizing each image, the leaf area was measured by automatically tracing the outline of the lamina with "Wand Tool" in ImageJ. When necessary, the original image was manually cropped (trimmed) to exclude the background outside the white clipboard to facilitate binarization (Figure 4; see Appendix A for details). Cropping was carefully performed so as to avoid changing the original resolution of the image. Except for cropping, no digital manipulations (adjustments for brightness, contrast, or resolution) were performed on the raw images prior to measurement. In cases where automatic binarization failed, the outline of the leaf lamina was manually selected by using the "Polygon Tool" in ImageJ.

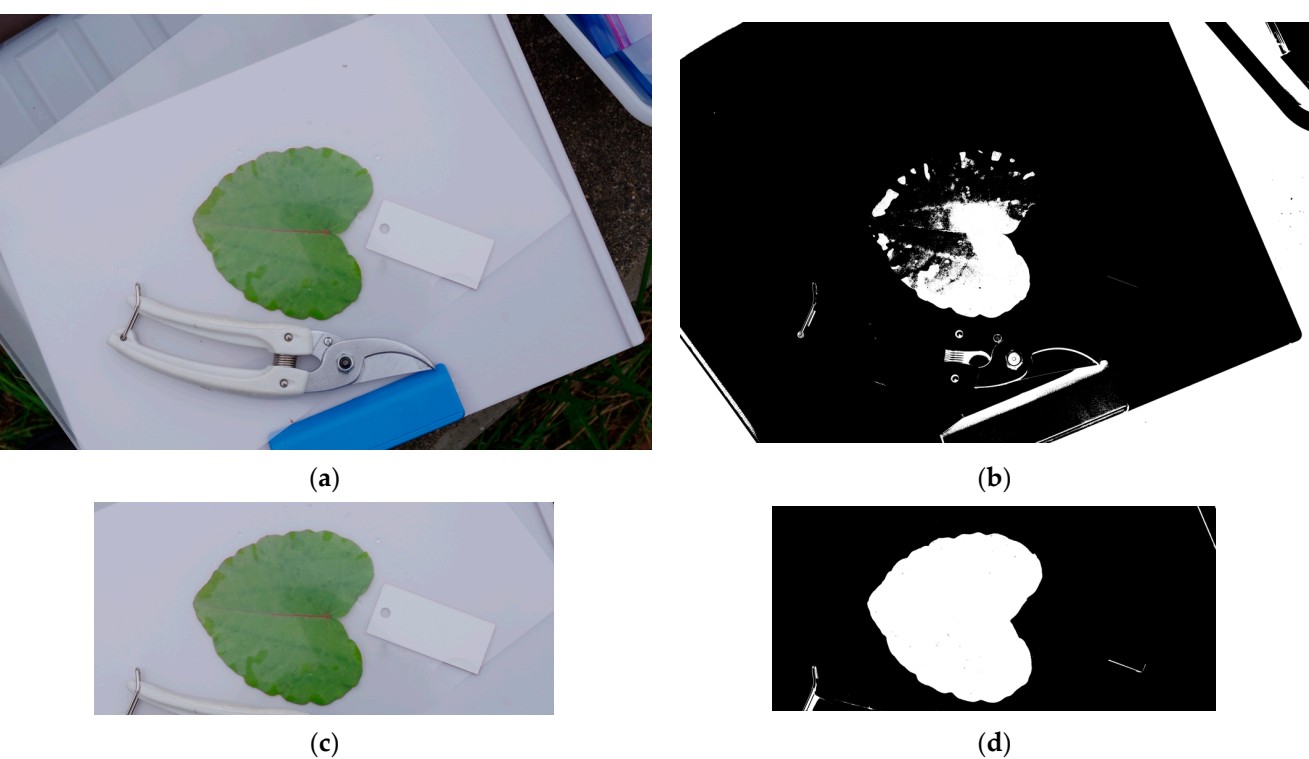

(**a**)  (**b**)

(**c**)  (**d**)

**Figure 4.** Image binarization for the automatic estimation of leaf area (see Appendix A for a manual of the measurement procedure using ImageJ software). A leaf lamina of *Fallopia sachalinensis* is shown. Automatic binarization was performed using the "Process → Binary → Make Binary" function in ImageJ. Panels (**a**–**d**) show the same photograph taken at different stages of the image analysis procedure. (**a**,**b**) A case in which binarization was not successful. (**c**,**d**) Binarization was successful after manually trimming the background outside the white clipboard. This photograph was taken in September 2022 by Kohei Koyama.

*2.5. Comparison of the Two Methods (Camera vs. Scanner)*

It is widely recognized that within each species, the area of an individual leaf ($A_{\mathrm{leaf}}$) is proportional to the product of its length ($L_{\mathrm{leaf}}$) and width ($W_{\mathrm{leaf}}$) [26]. The constant of proportionality in this relationship is referred to as the Montgomery parameter (*M*):

$$A_{\mathrm{leaf}} = M{\cdot}L_{\mathrm{leaf}}{\cdot}W_{\mathrm{leaf}} \tag{1}$$

To determine the accurate value of *M*, the conventional approach involves plotting a regression line of the relationship between the leaf area and the product of length and width. In this study, we instead calculated the Montgomery parameter for individual leaves ($M_{\mathrm{leaf}}$), which approximates the conventional *M*:

$$M_{\mathrm{leaf}} = A_{\mathrm{leaf}} / (L_{\mathrm{leaf}}{\cdot}W_{\mathrm{leaf}}) \tag{2}$$

We considered these values ($A_{\mathrm{leaf}}$ or $M_{\mathrm{leaf}}$) obtained with the scanner as the true values (denoted as $X_{\mathrm{scanner}}$) and compared them with those obtained with the camera ($X_{\mathrm{camera}}$). The absolute percentage error (*E*) was defined as the absolute value of the difference between these two values divided by the true value (Equation (3)), and the mean absolute percentage error was used as the representative of prediction performance.

$$E = |X_{\mathrm{camera}} - X_{\mathrm{scanner}}| / X_{\mathrm{scanner}} \times 100\% \tag{3}$$

*2.6. Estimation of the Montgomery Parameter from Five Images*

In addition to the five sample leaves mentioned above, we collected 50–55 leaves from each species. These were scanned, and the size parameters ($L_{\text{leaf}}$, $W_{\text{leaf}}$, and $A_{\text{leaf}}$) were determined using the same procedure described above. The Montgomery parameter ($M_{\text{regress}}$) was produced as the slope of a linear regression for each species; this $M_{\text{regress}}$ value was considered as the "true value." $M_{\text{regress}}$ was then compared with the mean of the five $M_{\text{leaf}}$ values calculated per species, which was obtained using digital camera photographs of five leaves as described above. For *Fallopia*, *Magnolia*, and *Prunus*, we skipped this procedure and instead used previously reported $M_{\text{regress}}$ values derived using the identical procedure [25]. The mean absolute percentage error (the mean of the five $M_{\text{leaf}}$ vs. $M_{\text{regress}}$) was calculated the same way as in Equation (3). Statistical analyses were performed using R v. 4.3.0 [82]. Linear regressions were performed using the *lm* function of R. Scatterplots were created using the packages *ggplot2* [83], *cowplot* [84], and *gridExtra* [85]. The dataset and R codes are available in the Supplementary Materials.

## 3. Results

We observed strong correlations ($R^2 > 0.99$) between the camera and scanner data. The regression slopes were both close to unity and the intercepts were close to zero (Figure 5). The mean absolute percentage error was 1.77% for individual leaf area ($A_{\text{leaf}}$) and 1.43% for the single-leaf Montgomery parameter ($M_{\text{leaf}}$).

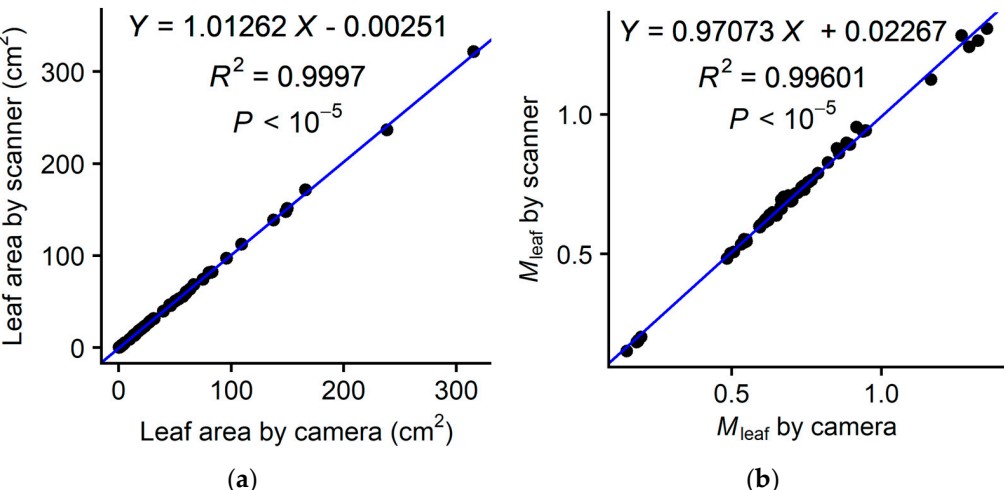

(**a**)  (**b**)

**Figure 5.** Comparison between the leaf area values obtained with a digital scanner and a digital camera. (**a**) Individual leaf area. (**b**) Single-leaf Montgomery parameter ($M_{\text{leaf}}$), which is defined as the lamina area divided by the product of lamina length and width (Equation (2)). A solid circle indicates one leaf ($n = 60$ leaves from the 12 species). The blue lines indicate ordinary least-squares regressions. The dataset is available in the Supplementary Materials.

We also observed a strong correlation between $M_{\text{regress}}$ and $M_{\text{leaf}}$ ($R^2 = 0.983$; Figure 6). The regression slope was close to unity (0.958), and the intercept was close to zero. The mean absolute percentage error was 4.20%. Note that the $M_{\text{leaf}}$ of *Hydrocotyle* were greater than one; this can be the case for species with protruding leaves (i.e., leaves in which the lamina extends below the base, *sensu* [26]).

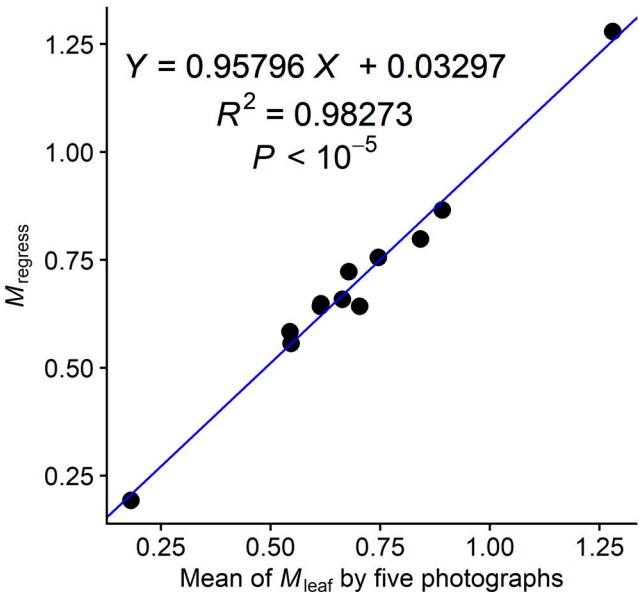

**Figure 6.** Comparison of the single-leaf Montgomery parameter of each species ($M_{\text{leaf}}$), calculated as the mean value of $M_{\text{leaf}}$ obtained from the digital camera photographs of five leaves, and the reference Montgomery parameter ($M_{\text{regress}}$), estimated using a standard, scanner-based method by scanning more than 50 leaves per species followed by a linear regression. A solid circle represents a single species ($n$ = 12). The blue line indicates ordinary least-squares regression. The dataset including $M_{\text{regress}}$ and $M_{\text{leaf}}$ of each species is available in the Supplementary Materials.

## 4. Discussion

Our findings suggest that the clear folder method can be used to estimate the area of leaves of varied shapes in situ with acceptable accuracy. Furthermore, the $M_{\text{leaf}}$ obtained from the five leaves may be used as an approximate value for obtaining the true Montgomery parameter ($M_{\text{regress}}$), although care should be taken for the estimation error.

The main advantage of the clear folder method is that it does not depend on the distance or angle between the leaf and camera. This is because the leaf area is calculated as the relative size between the leaf and scale (3 cm paper), both of which exist on the same flat plane on the clipboard. We found that this property greatly simplifies photographing in the field, because the leaves do not need to be fixed at a particular angle or distance from the camera. The only criterion required for the photograph is that the margin of the leaf lamina should be clearly distinguished from the background. As a result, the camera settings (exposure, illumination, resolution, weather, and time of day) do not have any significant effects on the results. This suggests that any inexpensive camera (including mobile phones) is suitable for this method. In addition, because this method does not require any large or expensive instruments, it can be implemented during fieldwork in forests, mountains, and farmlands. This simplicity and low-cost property is an advantage of this method over advanced 3D-techniques, which require specific camera setups, for example, fixed at a position in a laboratory [71–73], or complex image processing [66,76,77].

The present method does not rely on specific image analysis software. Therefore, after photographing the leaves, any automated image analysis software (e.g., Easy Leaf Area [43], LAMINA [86], LeafAnalyser [87], LeafJ [88], Black Spot [89], or pliman [38]) can be combined with this method to estimate the leaf area and Montgomery parameter. Deriving the Montgomery parameter using the clear folder method may also be useful for estimating the wing area of insects, which uses the same length-times-width equation [90]. However, as the present results are based on limited experimental conditions, further studies that include more species are needed before generalizing the applicability of this approach.

**Supplementary Materials:** The dataset and photographs are available online at: https://www.mdpi.com/article/10.3390/horticulturae9060709/s1.

**Funding:** This research was funded by the JSPS KAKENHI Grant Numbers 18K06406, 19H02987, and 23K05931.

**Data Availability Statement:** The dataset, R codes, and original high-resolution photographs are available as the Supplementary Materials.

**Conflicts of Interest:** The author declares no conflict of interest.

## Appendix A.

### *Appendix A.1. Outline*

This Appendix describes the basics of the leaf area measurement procedure for readers unfamiliar with ImageJ software. Digital images consist of pixels (or "dots"). Measuring leaf area (or any object, such as insect wing area) is equivalent to counting the number of pixels that comprise the object (Figure A1).

**Figure A1.** Outline of the area measurement procedure.

### *Appendix A.2. Taking a Picture*

Take a picture of an object (for example, a leaf) using a digital camera, mobile phone, or a digital scanner. When taking the photograph, the image should include a scale of known length (e.g., 3 cm). The details of photographing protocol are described in "Section 2.2. *Leaf Photography*" in the main text.

### *Appendix A.3. Opening the Image File with ImageJ*

1. ImageJ is a free software (in public domain) developed by Schneider et al. [81] that can be downloaded via its official page.
2. Launch ImageJ by double-clicking the ImageJ icon (Figure A2a). The "tool bar" (Figure A2b) will appear;
3. Drag and drop your image file (e.g., JPEG, PNG) onto the tool bar.

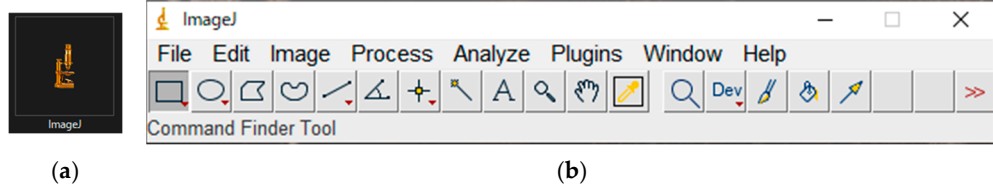

(**a**)  (**b**)

**Figure A2.** (**a**) ImageJ icon; (**b**) Tool bar.

## Appendix A.4. Measuring the Scale

1.  Select the "Straight tool" on the tool bar (Figure A3a);
2.  Left click the two endpoints of the 3 cm scale (Figure A3b). NOTE: Selection of the endpoints must be done precisely as this will affect the accuracy of the measurement;
3.  "Analyze" → "Measure" (or shortcut "Ctrl + M"); the "Results" window will appear (Figure A3c).

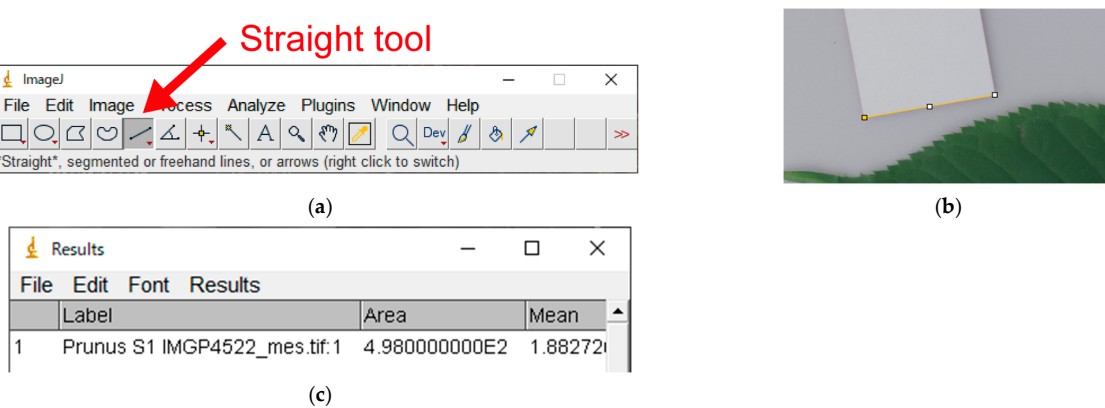

**Figure A3.** Measurement of the 3 cm scale. (**a**) Straight tool; (**b**) Selection of the endpoints of the 3 cm scale; (**c**) The Results window.

## Appendix A.5. Measuring Leaf Length and Width

Use the "Straight tool" to measure the leaf length ($L_{\text{leaf}}$) and width ($W_{\text{leaf}}$) (see Figure 3 in the main text) in the same way as the "Section A.4. *Measuring the Scale*" described above.

## Appendix A.6. Automatic Binarization Using ImageJ

1.  A binarized image consists of black and white pixels (no intermediate gray pixels). After binarization, the area of the leaf lamina (i.e., leaf blade) can be automatically measured using Image by counting the number of white (or black) dots.
2.  "Process" → "Binary" → "Make Binary" (Figure A4). NOTE: In ImageJ, the "Undo" command does not always work. Save the image at each step of the process (e.g., before and after binarization) using a different filename;
3.  If the automatic binarization process fails, see Section A.7 "*Image Cropping to Aid Automatic Binarization*" below.

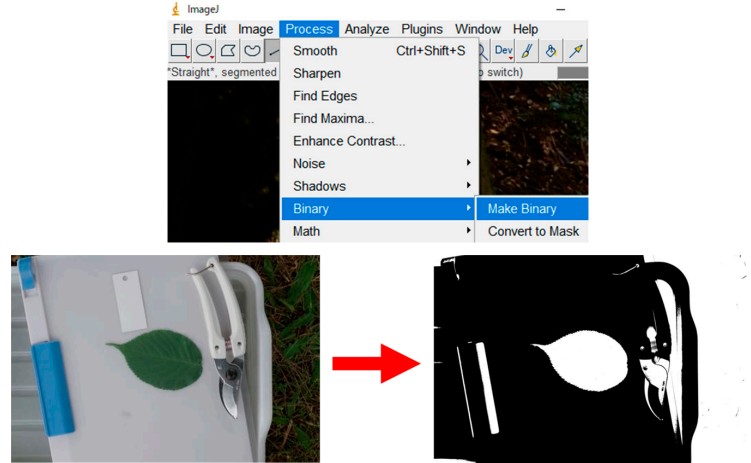

**Figure A4.** Binarization using ImageJ.

*Appendix A.7. Image Cropping to Aid Automatic Binarization (Optional)*

1.  Automatic binarization using ImageJ is not always successful. Cropping (trimming) the image often solves this problem (Figure A5a). This is because ImageJ automatically sets binarization threshold (unless it is manually specified) considering the entire image. Cropping background outside the white clipboard results in a "leaf vs. white background," which makes it easier for ImageJ to distinguish the leaf;

2.  "Rectangle tool" (Figure A5b) → Select the rectangle crop area (Figure A5c) →"Image" → "Crop" (Figure A5d);

3.  If binarization fails even after cropping, see Section A.12 "*Manual Leaf Selection with Polygon Selections*" below.

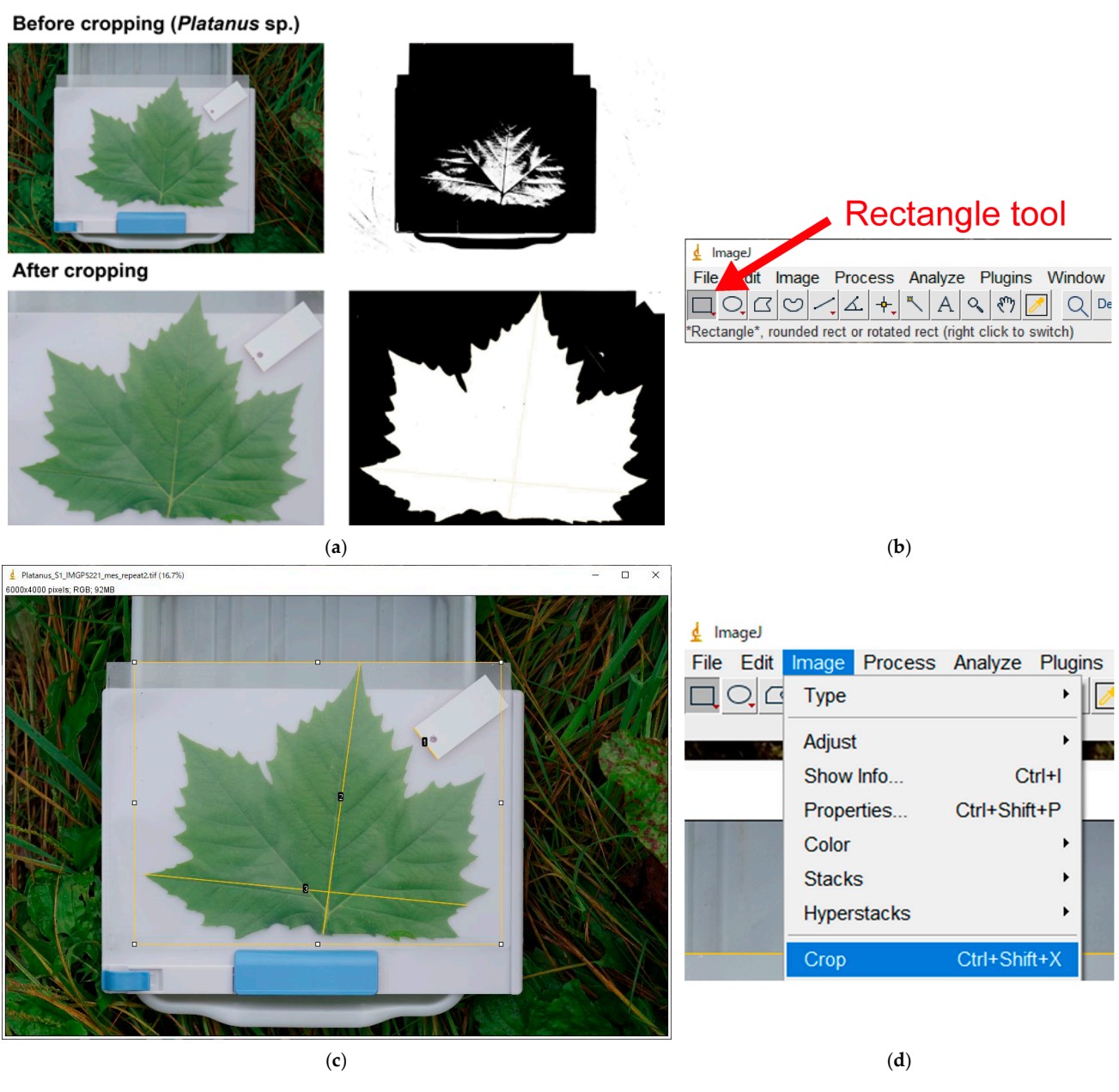

**Figure A5.** Image cropping to aid binarization. (**a**) Automatic binarization using ImageJ is not always successful. Cropping (trimming) the image often solves this problem; (**b**) Rectangle tool; (**c**) Selecting the rectangle crop area; (**d**) "Image" → "Crop" menu.

*Appendix A.8. Measuring the Leaf Area*

1. Select the "Wand (tracing) tool"(Figure A6a);
2. Activate the leaf silhouette by left clicking on it. If binarization is successful, ImageJ will automatically select the silhouette of the entire leaf lamina. Ensure that the circumference of the leaf lamina is correctly color-highlighted;
3. Press the "Ctrl + M" keys (or "Analyze" → "Measure"); the circumference of the lamina is now highlighted in light blue (the color may differ due to software settings) (Figure A6b). The results will be added as a new row in the "Results" window (Figure A6c). The area of the leaf, expressed as the number of pixels (not in $cm^2$), appears in the "Area" column in the last row in the Results window (i.e., the lamina silhouette consists of 1.376E6 = $1.376 \times 10^6$ = 1,376,000 pixels).

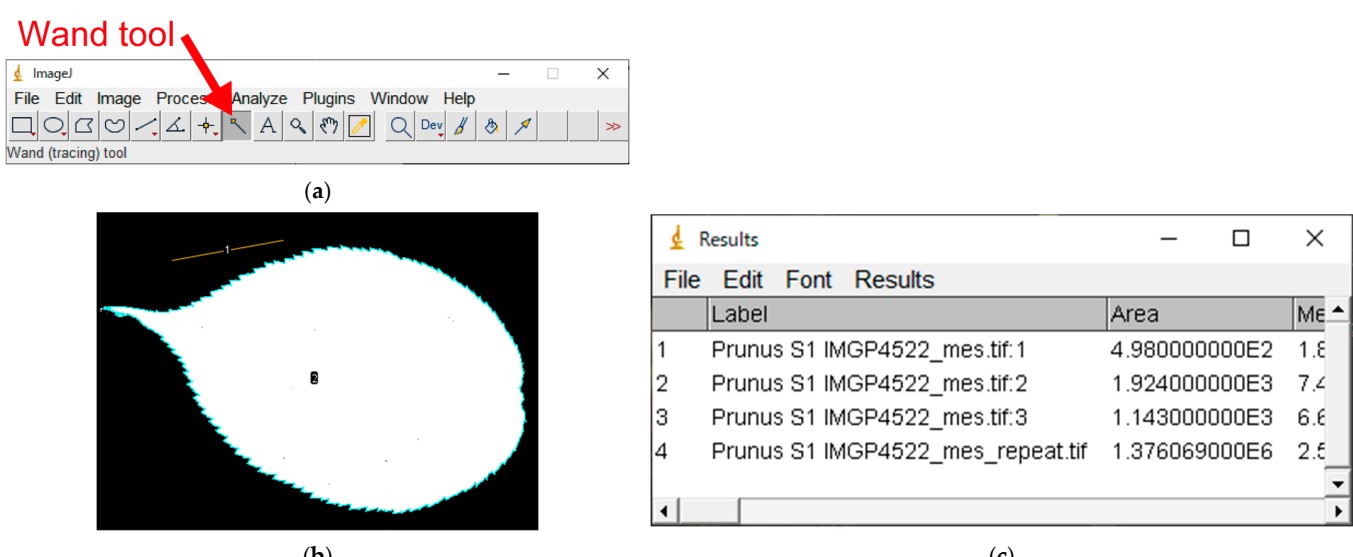

**Figure A6.** Measuring the leaf area. (**a**) Wand tool; (**b**) The circumference of the lamina highlighted in light blue; (**c**) The Results window.

*Appendix A.9. Exporting the Results as a CSV file*

Activate the "Results" window and save the results by "File" → "Save As" (Figure A7); a CSV file will be exported. Alternatively, you can copy and paste the results into Excel.

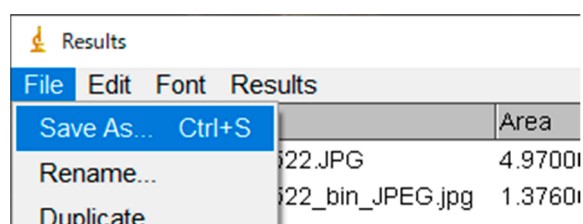

**Figure A7.** Exporting the results as a CSV file.

*Appendix A.10. Calculating the Resolution of the Image*

1. The resolution of an image will differ depending on the image file. If a camera is closer to a leaf, the same leaf will appear "larger" and thus contain more pixels in your image. We are interested in the actual leaf area (expressed in $cm^2$) not the number of pixels. Therefore, an image-specific conversion factor from pixels to $cm^2$ is needed. This conversion factor is called the resolution of the image and expressed as dots per inch (dpi).

2. A resolution of 200 dpi indicates that 200 dots (pixels) are placed within a line of 1 inch (2.54 cm) (Figure A8a). Hence, $200 \times 200 = 40{,}000$ dots are placed within a 2.54 cm $\times$ 2.54 cm square (Figure A8b).

3. In the Results window (or the exported CSV file), the first row presents the measurement results for the 3 cm scale (as we measured the scale first). The value in the "Length" column indicates how many pixels are placed on the distance of 3 cm on your image (Figure A8c).

4. Calculate the resolution of your image using the result for the 3 cm scale bar. For example, if the measured length of the 3 cm scale was 497 pixels (appearing as "4.967E2" (=496.7) in the "Results" window; Figure A8c), the resolution is $497/3 = 165.6$ dots per centimeter (=27,415 dots per cm$^2$). This corresponds to 420.6 dpi.

5. The actual leaf length and area are obtained by converting the units from pixels (dots) into cm (or cm$^2$): Leaf length = 1923 pixels/165.6 dots per cm = 11.62 cm, Leaf area = 1,376,069 pixels/27415 dots per cm$^2$ = 50.19 cm$^2$.

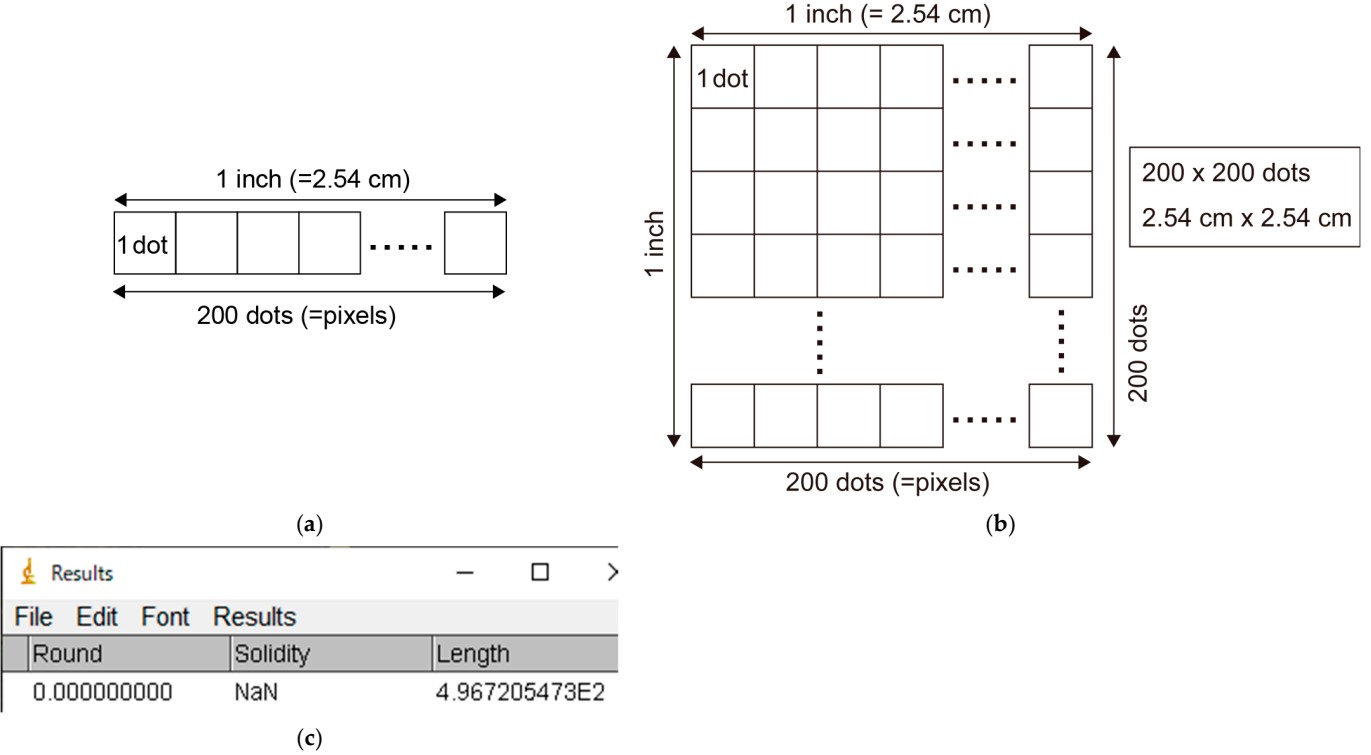

(a)

(b)

(c)

**Figure A8.** Calculating the resolution of the image. (**a**) A resolution of 200 dpi indicates that 200 dots (pixels) are placed within a line of 1 inch (2.54 cm). (**b**) Hence, $200 \times 200 = 40{,}000$ dots are placed within a 2.54 cm $\times$ 2.54 cm square. (**c**) The results showing the number of dots on the 3 cm scale. By using this value, you can calculate the resolution of your image.

*Appendix A.11. Measuring Leaf Area using Scanned Images*

1. Use white background (e.g., white paper) to facilitate binarization (Figure A9). Then, automatic binarization usually works well for scanned images. Ensure that you scan a scale bar (e.g., 3 cm) together with the leaves (Figure A9).

2. Remember the resolution (dpi) selected for scanning. Taking a screenshot of the scanner settings will save the dpi value together with the scanned images. This dpi value can be used to convert between units (dots into centimeters) without measuring the scale. Nevertheless, we recommend double-checking the dpi by measuring the scale bar even for scanned images. This is because the dpi of the image can be easily changed after editing the images.

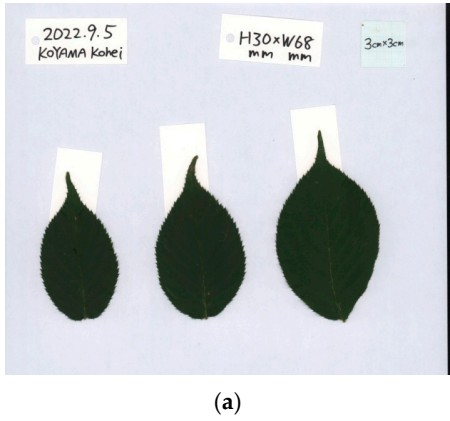 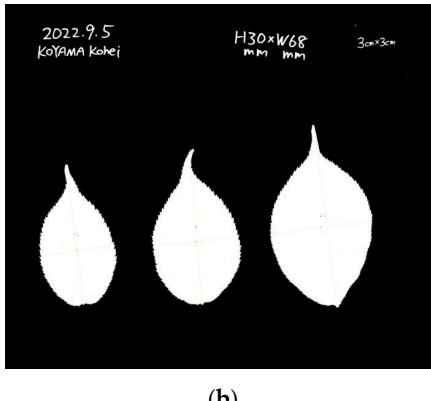

(**a**) (**b**)

**Figure A9.** (**a**) Scanned leaf laminas of Sargent's cherry (*Prunus sargentii* Rehder). The small paper cards placed at the tips of the laminas were to "stretch out" the lamina tips. We used large white paper as the background of the three leaves to facilitate binarization. (**b**) Binarized images.

*Appendix A.12. Manual Leaf Selection Using Polygon Selections (Optional)*

1. If binarization fails even after image cropping, you can measure the leaf area by manually tracing all edges of the leaf lamina: "Polygon selections" (Figure A10a) → manually select all edges of the leaf lamina (Figure A10b) → "Ctrl + M";
2. Although this is time-consuming, this generic method will work for any image in which the target object cannot be automatically separated from the background (e.g., when the object is not on a white background);

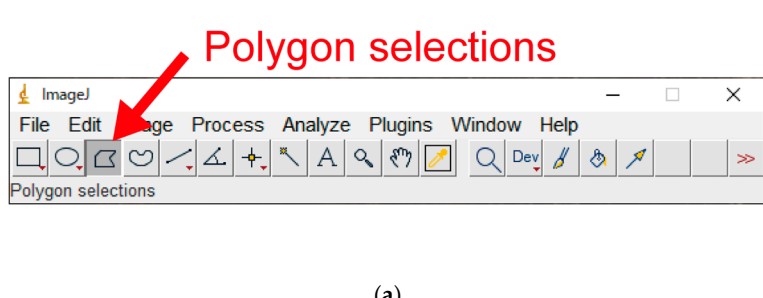 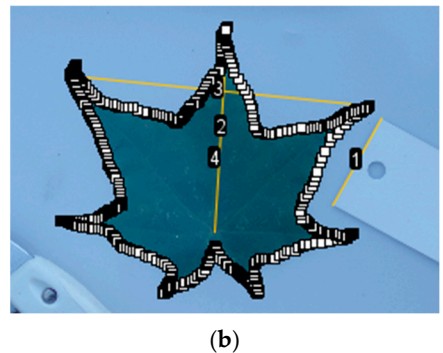

(**a**) (**b**)

**Figure A10.** (**a**) The "Polygon selections" tool in ImageJ. (**b**) Manual selection of the leaf lamina.

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
