# Peer review of "Leaf Area Estimation by Photographing Leaves Sandwiched between Transparent Clear File Folder Sheets"

_horticulturae, doi:10.3390/horticulturae9060709_

Round 1
Reviewer 1 Report
The author treat a methodological problem about leaf area estimate in the field with low-cost camera. They found a good agreement among methods.
Line 90 - Why Fallopia sachalinensis is in italic bold? Check also further in the text
Line 209-210 - Please add the functions used within each R package (at least the main one)
Lines 211- Please add some more detail to characterize better your relationships reported in Figures 4 and 5. Such as if intercept was statisticcally significant and if slope differed significantly from 1. The calculation of root mean square error (RMSE) estimate would also be appreciated.
Author Response
Please find the author’s reply to the Reviewer Report in the PDF file.

Reviewer 2 Report
This communication estimate leaf area by taking pictures of various leaves sandwiched between transparent clear file folder sheets. Authors claim that any inexpensive camera (including mobile phones) is suitable for this method and this method does not require any extensive or expensive instruments and uses only a camera, plastic sheets, and a clipboard, it can be implemented during fieldwork in forests, mountains, and farmlands. In order to prove the precision of this method they compared the data with the standard procedure and the results were pretty similar. The novelty is not extraordinary. However, I consider this research would be interesting for researchers as a new method of estimating the leaves area. The research objectives were clear and the methodology and research results were fully discussed. Therefore, I suggest accepting this communication.
- Page 249. Why references are taken to brackets?
- It would be adequate to process the pictures and show the length and width of the leaves that the software used to measure (Fig. 5).
Author Response

(The authors gave the same response as above.)

Reviewer 3 Report
The MS "Leaf area estimation by photographing leaves sandwiched be- 2 tween transparent clear file folder sheets" is quite useful for those who had problems to measure area of not flat leaves. The introduction is convincing, the results are clear. I only doubt the necessity of such a long reference list (122) for a communication. For instance, references on lines 165-166 do not seem to be needed.
Author Response

(The authors gave the same response as above.)

Reviewer 4 Report
1- In page 8, line 185, author refers to a "process->binary->make binary" process. Could you specify the software that was utilized to perform this task?
2- It is an interesting low cost proposal for leaf area measurement. However, i suggest to add an experiment setup diagram to clarify your prototype.
3- I insist that it is necessary to add labels to setup parts such as the mentioned 3cm scale.
4- it is necessary to specify if the leaf trimming process was performed manually of if there is an automatic object trimming algorithm. There are many algoritms to perform this.
5- In figure 2, there are many photographs about different leaves. However, I suggest to add an arrangement of rgb images ans it binarized counterpart in order to show how the environment around of the leave can affect the binarization process.
6- it is important to mention that low cost should be one of the main objectives of this proposal when it is compared to commercial leaf area measurement equipment.
Author Response

(The authors gave the same response as above.)
